# Exploring the Regulatory Effect of LPJZ-658 on Copper Deficiency Combined with Sugar-Induced MASLD in Middle-Aged Mice Based on Multi-Omics Analysis

**DOI:** 10.3390/nu16132010

**Published:** 2024-06-25

**Authors:** Chunhua Li, Ziqi Liu, Wei Wei, Chen Chen, Lichun Zhang, Yang Wang, Bo Zhou, Liming Liu, Xiao Li, Cuiqing Zhao

**Affiliations:** 1College of Animal Science and Technology, Jilin Agricultural Science and Technology University, Jilin City 132101, China; chunhuali2009@sina.com (C.L.); lzq11302024@126.com (Z.L.); chenchen20030101@126.com (C.C.); zlc88@126.com (L.Z.); aliuliming1984@126.com (L.L.); 2State Key Laboratory of Pathogen and Biosecurity, Beijing Institute of Microbiology and Epidemiology, Beijing 100071, China; weiwei_7776@hotmail.com; 3Jilin Ginseng Academy, Changchun University of Chinese Medicine, Changchun 130117, China; white-wing@163.com; 4Research Unit of Key Technologies for Prevention and Control of Virus Zoonoses, Changchun Veterinary Research Institute, Chinese Academy of Medical Sciences, Changchun 130122, China; hottank3210@126.com; 5Academician Workstation of Jilin Province, Changchun University of Chinese Medicine, Changchun 130117, China

**Keywords:** metabolic dysfunction-associated steatotic liver disease, copper deficiency, liver lipidomics, serum metabolomics, gut microbiota

## Abstract

Globally, metabolic dysfunction-associated steatotic liver disease (MASLD), previously termed nonalcoholic fatty liver disease (NAFLD), is one of the most common liver disorders and is strongly associated with copper deficiency. To explore the potential effects and mechanisms of *Lactiplantibacillus plantarum* LPJZ-658, copper deficiency combined with a high-sugar diet-induced MASLD mouse model was utilized in this study. We fed 40-week-old (middle-aged) male C57BL/6 mice a copper-deficient and high-sugar diet for 16 weeks (CuDS), with supplementary LPJZ-658 for the last 6 weeks (CuDS + LPJZ-658). In this study, we measured body weight, liver weight, and serum biochemical markers. Lipid accumulation, histology, lipidomics, and sphingolipid metabolism-related enzyme expression were investigated to analyze liver function. Untargeted metabolomics was used to analyze the serum and the composition and abundance of intestinal flora. In addition, the correlation between differential liver lipid profiles, serum metabolites, and gut flora at the genus level was measured. The results show that LPJZ-658 significantly improves abnormal liver function and hepatic steatosis. The lipidomics analyses and metabolic pathway analysis identified sphingolipid, retinol, and glycerophospholipid metabolism as the most relevant metabolic pathways that characterized liver lipid dysregulation in the CuDS group. Consistently, RT-qPCR analyses revealed that the enzymes catalyzing sphingolipid metabolism that were significantly upregulated in the CuDS group were downregulated by the LPJZ-658 treatment. In addition, the serum metabolomics results indicated that the linoleic acid, taurine and hypotaurine, and ascorbate and aldarate metabolism pathways were associated with CuDS-induced MASLD. Notably, we found that treatment with LPJZ-658 partially reversed the changes in the differential serum metabolites. Finally, LPJZ-658 effectively regulated intestinal flora abnormalities and was significantly correlated with differential hepatic lipid species and serum metabolites. In conclusion, we elucidated the function and potential mechanisms of LPJZ-658 in alleviating copper deficiency combined with sugar-induced middle-aged MASLD and hope this will provide possible treatment strategies for improving MASLD.

## 1. Introduction

Metabolic dysfunction-associated steatotic liver disease (MASLD; formerly non-alcohol-related fatty liver disease (NAFLD)) is the most common liver disease globally and causes the majority of liver-related morbidity and mortality [1,2]. The spectrum of MASLD ranges from simple steatosis and nonalcoholic steatohepatitis (NASH) with variable degrees of fibrosis to cirrhosis, which is a major risk factor for hepatocellular carcinoma [3]. In most patients, MASLD is associated with metabolic comorbidities such as obesity, type 2 diabetes mellitus (T2DM), or dyslipidemia, and it is often considered to be a hepatic manifestation of metabolic syndrome [4]. MASLD has become a global health problem, with its prevalence increasing significantly over time; the overall prevalence of MASLD is estimated to be 32.4% worldwide [5]. Generally, gender and age differences exist in MASLD patients. Specifically, the overall prevalence of MASLD and NASH is significantly higher in men than in women. More so than with gender, the prevalence of MASLD increases with age and is significantly higher among people aged 50 years and older than in people under 50 years [5,6]. Copper is an essential dietary trace element and is a structural and enzymatic co-factor for proteins involved in critical cellular processes, including iron metabolism, mitochondrial function, and redox homeostasis [7,8]. The liver is the central organ for copper homeostasis [9,10], and hepatic copper is a key modulator of lipid catabolism [11]. Inadequate copper intake is another important nutritional problem worldwide. There is growing evidence that copper deficiency (CuD) is associated with the development of MASLD [12,13,14]. Liver copper levels are lower in patients with MASLD compared with normal controls and individuals with other diseases [4]. Copper deficiency exacerbates hepatic fibrosis in bile duct-ligated rats, possibly mediated through reduced antioxidant defense and iron overload [15]. Notably, copper deficiency is relatively common in advanced cirrhosis and is associated with an increased risk of infection, a unique metabolic profile, and an increased risk of death before transplantation [16]. In addition, sugar intake further promotes the development of MASLD. Patients with MASLD have been shown to consume excessive amounts of fructose [17]. Thus, increased sugar intake and inadequate copper intake are two key risk factors for MASLD. It is notable that dietary fructose–copper interactions impair the copper status and exacerbate MASLD [18,19].

Probiotics have a long history of use and have attracted public interest in the course of promoting human intestinal health. The intake of probiotics has been suggested to enhance overall health and immunity. Studies have shown that probiotics can inhibit the growth and reproduction of pathogenic bacteria, regulate glucose and lipid metabolism, modulate the intestinal microbiota, enhance epithelial barrier function, and reduce intestinal inflammation [20]. In several studies involving humans and animals, probiotics have been reported to have the potential to ameliorate MASLD, as shown in the associated improvement in hepatic steatosis, inflammation, oxidative damage, and fat reduction [21,22,23]. Notably, our previous studies also identified that *Lactobacillus rhamnosus* GG (LGG) ameliorates MASLD by modulating intestinal barrier function and lipid metabolism [24,25,26,27]. However, few studies have focused on the role and mechanism of probiotics in copper deficiency-induced fatty liver disease.

*Lactiplantibacillus plantarum* LPJZ-658 is a probiotic strain newly isolated by our group [28]. In our previous study, we used metabolomics and gut microbiota analyses and found that LPJZ-658 mitigated the development of NASH by correcting the dysregulation of bile acid metabolism and modulating the gut microbiota [29]; thus, its potential role in improving MASLD was clarified.

In this study, we utilized serum metabolomics, liver lipidomics, and gut microbiome analysis to elucidate the potential role and mechanism of LPJZ-658 in ameliorating copper deficiency combined with sugar-induced MASLD in middle-aged mice.

## 2. Materials and Methods

### 2.1. Probiotic Strain

LPJZ-658 is a conserved strain grown in our laboratory. LPJZ-658 was cultured anaerobically in MRS broth (Beijing Solarbio Science & Technology Co., Ltd., Beijing, China) at 37 °C for 18–20 h.

### 2.2. Animal Experimental Design and Sample Collection

The 40-week-old male C57BL/6J mice in this study were randomly divided into three diet treatment groups: the CuA group, CuDS group, and CuDS + LPJZ-658 group. All the groups were fed ad libitum for 14 weeks; diets were based on a modified AIN93G formulation (Xiao Shu You Tai (Beijing) Biotechnology Co., Ltd., Beijing, China). The CuA group was fed a formula containing 17.8 mg/kg Cu and normal sterilized tap water. The CuDS group and CuDS + LPJZ-658 group were fed formulas containing a diet with less than 0.1 mg/kg Cu and sugar solution consisting of glucose (18.9 g/L) and fructose (23.1 g/L) (Sigma-Aldrich, St. Louis, MO, USA). In addition, in the last 6 weeks of the experiment, the CuDS + LPJZ-658 group was orally administered daily with 1 × 10^9^ CFU/day per mouse. After all experiments, the mice were fasted for 12 h and then euthanized for serum, liver, and cecal-content collection. Mice serum samples were processed for lipid profiles and metabolomics analysis. Liver samples were processed for histology, lipid profiles, lipidomic analysis, and gene expression analysis. Cecal contents were collected for gut microbiota analysis. For the animal experiment, n = 6–8 per group. Experiments were repeated three times.

### 2.3. Blood Lipid Profiles and ALT/AST Assays

Whole blood was centrifuged to separate serum. Alanine Aminotransferase Assay Kit and Aspartate Aminotransferase Assay Kit (Nanjing Jiancheng Institute of Bioengineering, Nanjing, China) were used to measure ALT and AST activity in serum samples and were utilized according to the manufacturer’s instructions. Serum lipid profiles, including triglyceride (TG), total cholesterol (T-CHO), high-density lipoprotein (HDL), low-density lipoprotein (LDL), and nonesterified fatty acid (NEFA), are measured using their corresponding assay kits (Nanjing Jiancheng Institute of Bioengineering, Nanjing, China).

### 2.4. Histological Analysis

Liver samples were collected, fixed with 4% paraformaldehyde overnight, embedded into paraffin wax blocks, cut into 5-µm-thick sections, and stained with hematoxylin and eosin (H&E) (Wuhan Servicebio Technology Co., Ltd., Wuhan, China).

### 2.5. Liver Lipid Profiles Assay

After homogenization and centrifugation (2500 rpm, 15 min, 4 °C) of liver tissue in ice-cold sodium chloride buffer, the supernatant was subjected to the determination of hepatic TG and T-CHO content using commercially available kits (Nanjing Jiancheng Institute of Bioengineering, Nanjing, China).

### 2.6. Statistical Analysis

Statistical significance was assessed using GraphPad Prism 7.0 software (GraphPad Software Inc., La Jolla, CA, USA). Data are presented as the means ± SEM. Differences between multiple groups were analyzed using one-way ANOVA, and a *p*-value < 0.05 was defined as the level of significance. Significance is noted as * *p* < 0.05, ** *p* < 0.01, and *** *p* < 0.001. The correlation analysis between gut microbiota, serum metabolites, and liver lipidomics was investigated using Cytoscape V3.7.1 software.

Additional methods are described in Appendix A.

## 3. Results

### 3.1. Effect of LPJZ-658 on Liver, Body Weight, and Blood Metabolites in MASLD Mice

To examine the effects of LPJZ-658 on hepatic injury, we used a copper deficiency diet and combined it with a sugar solution-induced mouse model for MASLD. As shown in Figure 1A–C, there were no significant differences in the body weight, liver/body weight, and eWAT (epididymal white adipose tissue)-to-body weight ratios between any of the groups. The metabolic index of the mice is shown in Figure 1D. In comparison to the CuA group, the serum levels of T-CHO, LDL, and NEFA were significantly elevated, and the HDL levels were significantly lower in both the CuDS group and the CuDS + LPJZ-658 group of mice. Unexpectedly, LPJZ-658 did not affect CuDS-induced serum dyslipidemia. We next evaluated liver injury by assessing the plasma levels of the hepatic enzymes AST and ALT. As shown in Figure 1E, the CuDS and CuDS + LPJZ-658 treatment did not affect the plasma ALT level. However, the serum AST levels were significantly increased in the CuDS group and markedly decreased in the CuDS + LPJZ-658 group. These results suggested that the LPJZ-658 intervention did not significantly improve the levels of relevant lipids in the peripheral blood of the mice compared to the CuDS group. However, LPJZ-658 treatment might effectively prevent liver injury in Cu deficiency combined with sugar-induced MASLD in mice.

### 3.2. Effect of LPJZ-658 on Liver Steatosis in MASLD Mice

The accumulation of fat in the liver is one of the characteristics of liver injury in MASLD. To evaluate the effect of LPJZ-658 on Cu deficiency combined with sugar-induced hepatic lipid accumulation, we subsequently conducted a histopathological examination of the liver. Regarding the H&E staining, the CuDS group demonstrated significantly enhanced formation of hepatic lipid droplets compared with the CuA group, and this decreased in the CuDS + LPJZ-658 group (Figure 2A). As expected, the LPJZ-658 treatment decreased the CuDS group-induced intrahepatic TG levels (Figure 2B). To further investigate whether liver lipid metabolism is associated with the biological activity of LPJZ-658, we analyzed differences in the lipid composition in mice liver using lipidomics methods.

The positive and negative ion current plots show an acceptable peak shape and relatively uniform distribution under the assay conditions of this experiment (Appendix A). A principal component analysis (PCA) was performed on the liver content samples of each group. As shown in Figure 2C,D, the samples in the CuA group were significantly distinct from those of the CuDS group, indicating that copper deficiency combined with sugar-induced significant lipid metabolite changes in the liver. The CuDS + LPJZ-658 group tended to separate from the CuDS group and partially overlap with the CuA groups, indicating that the disordered hepatic lipid metabolism was partially reversed after the LPJZ-658 intervention. Therefore, this method is reliable, and the results are credible. The OPLS-DA analysis and volcano plots of the differential metabolites in the dual-ion mode of the CuA and CuDS groups are shown in Figure 2E–H. According to the criteria of an FC of > 1.2 or < 0.83 and a VIP of > 1, 76 significantly different lipid species were identified in the CuDS vs. CuA groups (Appendix A). Taking the ion at RT 2.45 min *m*/*z* 566.3469 as an example, the MS spectrum in Figure 2I showed that the ion at *m*/*z* 566.3469 belonged to [M-H]^−^. As shown in Figure 2J, the compound produced three fragment ions, including *m*/*z* 506.3239, 281.2506, and 242.0799. The compound was identified as LysoPC (18:1(9Z)/0:0) after searching the LIPID MAPS database. To clarify the potential metabolic pathways of the 76 significantly differential lipid species identified by the CuDS/CuA comparison, a pathway-enrichment analysis was performed. As shown in Figure 2K, the results indicate that these lipid species mainly affect the sphingolipid, retinol, and glycerophospholipid metabolism pathways. These differential lipid species were plotted as heat maps to provide a more comprehensive picture of the differences between the three groups (Figure 2L). In addition, six significantly reversed differential lipid species between the CuDS + LPJZ-658/CuDS groups were compared (Appendix A). To validate the analysis of the lipidomic metabolites, we performed qRT-PCR to measure the alteration in the mRNA levels of the sphingolipid metabolic-related enzymes. Figure 2M shows that sphk2, Cers2, Sgms1, Smpd1, Smpd2, Smpd3, Smpd4, and Spt1c2 were significantly upregulated in the livers of the CuDS group mice. Notably, the expression levels of sphk2, Cers2, Cers4, Smpd1, Smpd2, and Spt1c2 were significantly downregulated upon LPJZ-658 treatment. These findings suggested that dietary marginal copper deficiency and fructose feeding interact to exacerbate hepatic fat accumulation, and treatment with LPJZ-658 might effectively prevent hepatic dyslipidemia features in Cu deficiency combined with sugar-induced MASLD in mice.

### 3.3. Effect of LPJZ-658 on Serum Metabolome Profile in MASLD Mice

UHPLC-Q-Orbitrap/MS was used to evaluate the serum extracts, and Appendix A displays the base peak chromatograms (BPCs) of the various groups. The peaks differed significantly in terms of the peak intensity and retention time. A multivariate statistical analysis was necessary for the identification of the metabolites due to the large number of ions present in each chromatogram. Following the generation of the Progenesis QI data, datasets, including the sample information, RT *m*/*z* values, and peak intensities, were generated for statistical analysis, and PCA models were constructed to visualize the classification patterns. Scoring plots were created based on the data obtained by LC-MS from serum extracts in both the positive and negative ion modes (Figure 3A,B). We can observe a clear difference between the CuA group and the CuDS group. After the treatment with LPJZ-658, the points in the CuDS + LPJZ-6518 group show separation from the CuDS group. However, there is still a partial crossover, suggesting that the LPJZ-658 treatment could partially restore the metabolic changes in the CuDS group. The OPLS-DA model-derived VIP > 1, *p* < 0.05 in one-way ANOVA, and FC > 1.2 or FC < 0.83 were used as the criteria for differentially expressed metabolites (Figure 3C–F). We identified 33 metabolites showing significant differences between the CuA and CuDS groups (Appendix A). The KEGG pathway-enrichment analysis showed that these metabolites affect multiple metabolic pathways (Figure 3G), including linoleic acid metabolism, taurine and hypotaurine metabolism, ascorbate and aldarate metabolism, etc. In addition, as shown in Figure 3H and Appendix A, the LPJZ-658 treatment significantly reversed seven serum metabolites compared to the CuDS group. These findings demonstrated that LPJZ-658 supplementation might mitigate the metabolic changes associated with Cu deficiency combined with sugar-induced MASLD in mice by adjusting these pathways.

### 3.4. Effect of LPJZ-658 on Gut Microbiota Composition in MASLD Mice

To elucidate the underlying mechanisms of LPJZ-658 in improving MASLD, we investigated its effects on the gut microbial composition. As shown in Figure 4A, a Venn diagram of the OTUs was generated. Furthermore, the OTU rarefaction curves tended to be flat, suggesting a credible sequencing result (Figure 4B). No significant differences were observed in the alpha diversity indices, including the Simpson, Chao 1, and Shannon indices, as presented in Figure 4C. The principal coordinate analysis (PCoA) results are shown in Figure 4D, representing the beta diversity, reflecting the changes in the gut microbiota composition between each group. The important microorganisms for improving the gut microbiota of LPJZ-658 were determined by a linear discriminant analysis effect size (LEfSe) analysis to be sets of bacteria with linear discriminant analysis (LDA) values greater than 4, which were considered to be significantly different. The LDA score distribution map and cladogram are shown in Figure 4E. As shown in Appendix A, a total of 13 bacterial genera in the cecal-content samples were detected as being significantly different by the LEfSe, of which the CuA group accounted for only one (Dubosiella). Eight different genera, within two phyla (Proteobacteria and unidentified_Bacteria), three classes (Desulfovibrionia, Gammaproteobacteria, and Alphaproteobacteria), two orders (Desulfovibrionales and Pseudomonadales), and one family (Desulfovibrionaceae) were found in the CuDS group. In the CuDS + LPJZ-658 group, four bacterial taxa were found within one order (Erysipelotrichales), one family (Erysipelotrichaceae), one genus (*Faecalibaculum*), and one species (*Faecalibaculum_rodentium*).

At the phylum level, Firmicutes, Bacteroidetes, and Verrucomicrobiota were the predominant phyla of microbiota in the cecal contents (Figure 4F). As shown in Figure 4G, the phylum-level analysis demonstrated that Cu deficiency combined with sugar-induced MASLD in the mice showed a significantly increased relative abundance of the Proteobacteria phylum, which was significantly reversed by the supplementation with LPJZ-658. In addition, a significant increase in the relative abundance of Firmicutes was observed in the CuDS + LPJZ-658 group compared to the CuDS group.

In addition, Figure 4H displays the relative abundance of the microbiota in the mouse cecal contents at the genus level (top 35) and the remarkable differences in the microorganisms, as shown in Figure 4I. The results showed that the relative abundances of Ruminococcus_torques_group were significantly increased in the CuDS group compared to the CuA group (Figure 4I), while LPJZ-658 supplementation significantly decreased them. In addition, *Faecalibaculum* was significantly elevated and Prevotellacea_UCG-001, [Eubacterium]_fissicatena_group was significantly decreased in the CuDS + LPJZ-658 group compared to the CuDS group. These results suggest that LPJZ-658 administration induces changes in the gut microbial composition, which may have beneficial effects on alleviating Cu deficiency with combined sugar-induced MASLD in mice.

### 3.5. Correlation Analysis of the Gut Microbiota–Serum Metabolites–Liver Lipidomic Axis

To determine whether the LPJZ-658-induced changes in the liver lipid metabolome and serum metabolome were associated with the effects of the gut microbiota, we selected the differential metabolites and genus-level colony abundance among the three groups for a Spearman correlation analysis. We found significant correlations between the LPJZ-658-induced change in lipid metabolomics, serum metabolomics metabolites, and genus-level gut microbiota abundance. As shown in Figure 5A,B, the liver lipid metabolomics PC (18:0/18:1) was significantly positively correlated with the serum 2-Butoxyethanol, and the PC (18:0/22:4) was significantly positively correlated with the serum Pantothenic acid, Sphingosine 1-phosphate, and 2-Butoxyethanol. Furthermore, SM (d17:1/24:1) and PG (22:6/22:6) were significantly negatively correlated with the serum Pantothenic acid, Sphingosine 1-phosphate, and Citrulline, and the gut abundance of [Ruminococcus]_torques_group. In addition, PG (22:6/20:3) was significantly positively correlated with the serum Pantothenic acid, Sphingosine 1-phosphate, Citrulline, and 2-Butoxyethano, and significantly negatively correlated with the gut abundance of Prevotellaceae_UCG-001. The serum metabolomics metabolite Citric acid showed negative correlations with the serum Sphingosine 1-phosphate and Citrulline and gut Prevotellaceae_UCG-001 abundance. The serum metabolomics metabolite Sphingosine 1-phosphate showed negative correlations with the serum 2-Butoxyethanol and gut Prevotellaceae_UCG-001 abundance. In addition, the serum Citrulline was positively correlated with the gut abundance of *Faecalibaculum* and negatively correlated with [Ruminococcus]_torques_group.

## 4. Discussion

MASLD is characterized by abnormal lipid metabolism, which results in lipid accumulation in hepatocytes [2]. However, there has been a lack of consensus on the specific mechanism responsible for this until now. Copper is an important dietary trace element that has an essential role in mammalian physiology [30]. Western-style diets are low in copper levels and high in fructose. In particular, given the central role of the liver in maintaining Cu metabolism, hepatic copper deficiency is strongly associated with metabolic diseases, including diabetes mellitus, metabolic syndrome, and MASLD [14,31,32]. Studies of rodent models have found that low dietary copper increases lipid accumulation in the liver [14] and induces dyslipidemia [33,34]. In recent years, many probiotics have been shown to prevent and treat MASLD [35,36,37,38]. LPJZ-658 is a strain of Lactiplantibacillus plantarum isolated by our group. Our practice has confirmed that LPJZ-658 improves lipid metabolism in late-laying laying hens and broiler chickens [39]. It is important to note that we found that LPJZ-658 ameliorated hepatic steatosis, inflammation, and fibrosis in NASH model mice by mediating bile acid metabolism and the gut microbiota [29], which suggests that LPJZ-658 has the potential to improve MASLD. In this study, copper deficiency combined with a high-sugar-diet-induced MASLD mouse model was used to explore the therapeutic mechanism of LPJZ-658 in MASLD-combined lipidomics, metabolomics, and gut microbiota strategies.

The present study is similar to previous studies utilizing copper deficiency combined with fructose-induced MASLD [13]; the lipid profiles were significantly disturbed in the CuDS group (Figure 1D). To our surprise, the treatment with LPJZ-658 did not improve this profile. We next evaluated liver injury by assessing the serum levels of hepatic enzymes; the AST levels were significantly increased in the CuDS group. Importantly, the elevated serum ALT levels were attenuated by the LPJZ-658 treatment (Figure 1E). Prior work suggests that low Cu enhances MASLD severity [14,40], while metabolic disorders and their toxic lipid hoarding products are considered central to the development of MASLD [41]. Here, we found that CuDS resulted in moderate hepatosteatosis through histology H&E staining and that LPJZ-658 supplementation alleviated hepatic steatosis (Figure 2A). Consistent with liver histology, the quantitative evaluations indicated that LPJZ-658 can also improve CuDS-induced hepatic steatosis by reducing the TG levels (Figure 2B).

Even though TGs are primarily associated with hepatic pathologies, other lipid fractions seem to be associated with the development and severity of MASLD. Therefore, in the present study, we investigated the effect of CuDS alone or after treatment with LPJZ-658 administration on hepatic steatosis, using untargeted lipidomic analyses to understand changes in the hepatic lipid profiles. The data analyses identified 76 differential lipid species associated with the comparison between CuDS and CuA (Appendix A). These 76 differential lipid species are mainly involved in the sphingolipid, retinol, and glycerophospholipid metabolism pathways (Figure 2K). Among them, sphingolipid metabolism attracted our attention. The sphingolipid and glycerolipid metabolic pathways share many common intermediates [42,43], and there is a strong correlation between the triglyceride and sphingolipid contents in the liver. The sphingolipid levels are involved in regulating major biological processes [44,45]. The dysregulation of sphingolipid metabolism has been found to contribute to the development of MASLD. The sphingolipid family includes ceramides, sphingosines, glycosphingolipids, and sphingomyelins [46]. As a critical bioactive lipid, ceramide is considered to be a metabolic hub as it lies at the crossroads of multiple metabolic pathways in mammals [47]. A previous study found a significant increase in the ceramide content in the liver of rats fed with a high-fat diet (HFD) [48]. Moreover, MASLD is associated with the increased expression of genes involved in the three different pathways leading to ceramide synthesis, including de novo synthesis, sphingomyelin hydrolysis, and the salvage pathway [48,49].

Ceramides can be directly synthesized from sphingosines and acyl-CoAs expressed by six distinct genes (Cers1–6) through a salvage pathway. Specifically, CerS and Cers2, which synthesize long-chain ceramides, are the most prominent and are mainly expressed in the liver [50,51]. The enzymatic breakdown of sphingomyelin by sphingomyelinase (SMase, Smpd) is performed by the Cer-synthesizing enzyme that catalyzes the hydrolysis of sphingomyelin to Cer [52,53]. The deletion of Smpd improves MASLD progression by protecting against endoplasmic reticulum stress [54]. There is further evidence that ceramides can also be generated from the degradation of complex sphingolipids, including glycerophospholipids and sphingomyelins. Next, ceramide can be converted by neutral ceramidase to a biologically active signaling molecule called Sphingosine (Sph), which can ultimately be phosphorylated by Sph kinase (SphK) to Sph-1-phosphate (S1P). S1P represents one of the final metabolites produced by sphingolipid metabolism and is a key biologically active sphingolipid involved in inflammation, cell survival, and migration [55,56]. By dissecting the dysregulation of sphingolipid metabolic enzymes at the mRNA level, we found that CuDS dramatically upregulated Sphk2, Cers2, Sgms1, Smpd1–4, and Spt1c2 in livers (Figure 2M). Importantly, elevated mRNA levels of Sphk2, Cers2, Smpd1, Smpd2, and Spt1c2 were attenuated by LPJZ-658, suggesting the LPJZ-658 is important in alleviating CuDS-induced disorders of sphingomyelin metabolism.

In MASLD, the phenotype is complex and dynamic as a result of multiple interactions between genetic and environmental factors. Metabolites can act as regulators of phenotypes. Metabolites present in blood provide rich information about individual physiological states. Many serum biomarkers have been reported to differentiate between MASLD and healthy individuals [57]. Therefore, metabolomics may provide more insights and clues for the present study on the development and progression of copper deficiency combined with sugar-induced MASLD and the mechanism of action of LPJZ-658. In our previous studies, LPJZ-658 has been found to affect multiple metabolic pathways, especially bile acid biosynthesis [29].

In this study, we examined the changes in metabolites between the three groups using untargeted metabolomics. A clear separation between the CuA and CuDS groups was observed by both ESI+ and ESI− in the PCA and OPLS-DA analyses, suggesting that CuDS treatment has a significant effect on metabolism in mice. We found 33 differential metabolites between the CuA and CuDS groups that mainly affected metabolic pathways, including linoleic acid metabolism, taurine and hypotaurine metabolism, and ascorbate and aldarate metabolism, which is strongly associated with MASLD development [58,59,60,61]. Furthermore, LPJZ-658 treatment reversed 6 of these 33 differential metabolites. In conclusion, these data demonstrated that supplementation with LPJZ-658 may improve MASLD by regulating multiple metabolic pathways.

The gut microbiota is an important bridge to systemic metabolism. The gut microbial imbalance may lead to the dysfunction of the host’s machinery, which contributes to the pathogenesis of MASLD [62]. To elucidate the mechanisms of LPJZ-658, we investigated the changes in the composition of the cecal-content microbiota. We analyzed the relative abundance of bacteria at the phylum and genus levels, respectively. At the phylum level, although the two dominant phyla, Bacteroidetes and Firmicute, did not change significantly in the CuDS group, we found that the abundance of Proteobacteria increased significantly, which is consistent with clinical findings [63]. The rise in the Proteobacteria phylum proportion was the most prominent alteration in the gut–liver axis-induced hepatic steatosis in high-fructose-induced MASLD [64]. Remarkably, we found a marked reduction in the Proteobacteria abundance in the CuDS + LPJZ-658 group (Figure 4G). Moreover, at the genus level, the CuDS group was characterized by a significantly higher relative abundance of Ruminococcus_torques_group compared with the CuA group. Noteworthily, LPJZ-658 supplementation significantly changed the gut microbiota composition to varying degrees. The CuDS + LPJZ-658 group displayed a significantly increased abundance of *Faecalibaculum* and a decreased relative abundance of Prevotellaceae_UCG-001, Ruminococcus_torques_group, and Eubacterium fissicatena group (Figure 4I). In the gut of patients and animals with MASLD, the abundance of Ruminococcus_torques_group is increased [65]. Interestingly, multiple studies have found that the intestinal Ruminococcus_torques_group was crucial in lipid deposition and hepatic steatosis in poultry [66,67,68]. In the present study, the reduction in Ruminococcus_torques_group abundance after the LPJZ-658 treatment may be involved in the regulation of hepatic lipid metabolism in MASLD mice. The genus Parabacteroides has been reported to be associated with obesity, insulin resistance, and inflammation in HFD-fed mice [69]. Although some strains of Prevotellaceae seem to be inflammatory pathogens, they play a critical role in immune regulation [70]. In the present study, we observed that the abundance of Prevotellaceae_UCG-001 decreased in the CuDS + LPJZ-658 group, suggesting that LPJZ-658 promoted the abundance of microbes regulating immunity. *Faecalibaculum* may mediate these beneficial effects through carbohydrate and energy metabolism pathways in rodents [71]. In our study, the LPJZ-658-induced increase in *Faecalibaculum* may favorably influence the function of energy metabolism and improve nutrient absorption and metabolic processes in the intestine. The Eubacterium fissicatena group is highly correlated with obesity and obesity-related metabolic disorders in hosts [72]. Another notable finding indicated that the proportion of the Eubacterium fissicatena group was significantly higher in MASLD mice [73,74]. Although CuDS did not elevate the abundance of the Eubacterium fissicatena group in the present study, LPJZ-658 significantly reduced its abundance. The above studies revealed the beneficial role of LPJZ-658 in regulating the intestinal flora. Therefore, the above studies revealed that LPJZ-658 may alleviate CuDS-induced hepatic steatosis through the modulation of the gut microbiota composition. More importantly, the results of the Pearson correlation showed a significant correlation between the differential liver lipid profiles, serum metabolites, and gut flora at the genus level (Figure 5).

## 5. Conclusions

In conclusion, this study has shown that copper deficiency combined with sugar causes liver steatosis and gut dysbiosis. Through a multi-omics analysis, we demonstrate that LPJZ-658 supplementation improves abnormalities of hepatic lipid metabolism significantly in middle-aged MASLD mice.

## Figures and Tables

**Figure 1 nutrients-16-02010-f001:**
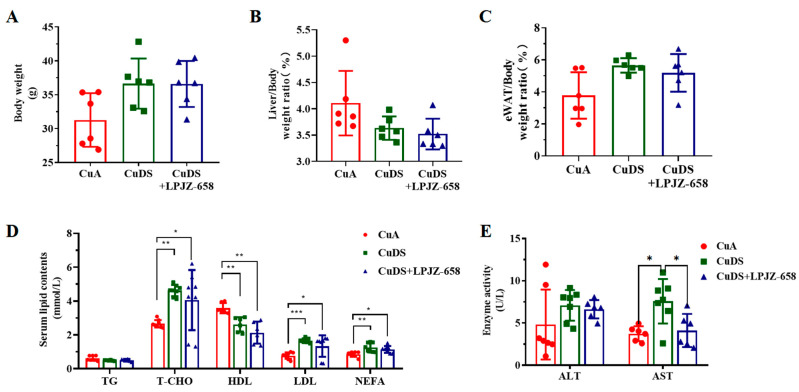
Effect of LPJZ-658 on body weight, serum lipid homeostasis, and liver injury in MASLD mice. (**A**) Body weight. (**B**) Liver/body weight ratio. (**C**) eWAT/body weight ratio. (**D**) Serum levels of TG, T-CHO, HDL, LDL and NEFA. (**E**) Serum levels of ALT and AST. eWAT: epididymal white adipose tissue, TG: triglyceride, T-CHO: total cholesterol, HDL: high-density lipoprotein, LDL: low-density lipoprotein, NEFA: nonesterified fatty acid, ALT: Alanine Aminotransferase, AST: Aspartate Aminotransferase. * *p* < 0.05, ** *p* < 0.01, and *** *p* < 0.001.

**Figure 2 nutrients-16-02010-f002:**
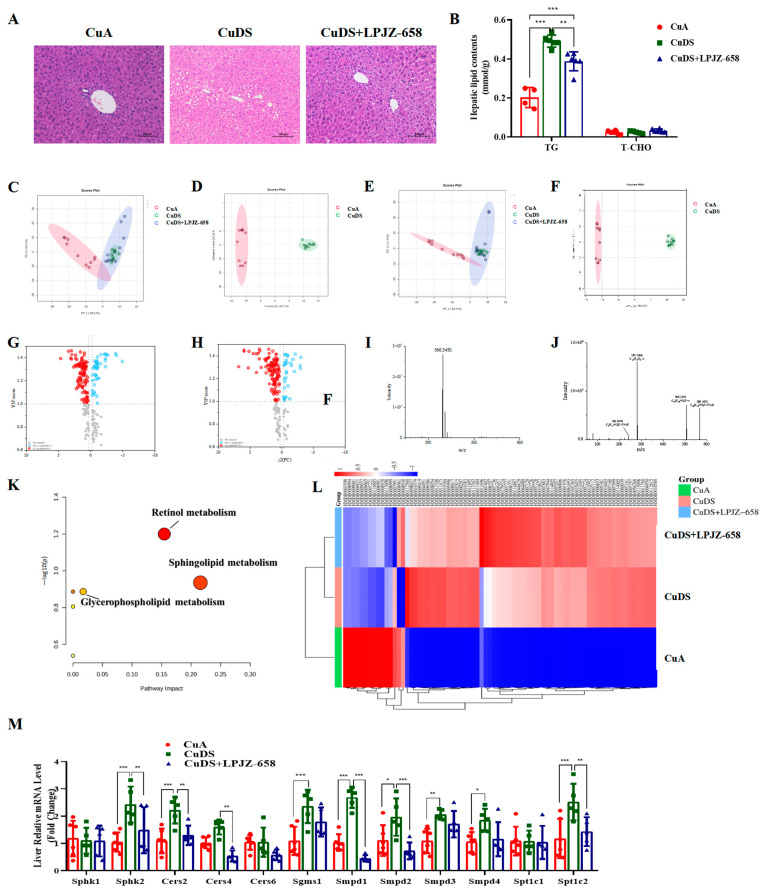
Effect of LPJZ-658 on hepatic steatosis and lipidomics profile in MASLD mice. (**A**) Representative images of H&E staining of liver tissue sections (scale bars: 100 µm). (**B**) Hepatic TG and T-CHO levels. PCA score plot of the annotated metabolites ((**C**) ESI+ and (**E**) ESI−). OPLS-DA analysis of identified metabolites ((**D**) ESI+ and (**F**) ESI−). Volcano plots of identified metabolites ((**G**) ESI+ and (**H**) ESI−). Mass spectra of the LysoPC(18:1(9Z)/0:0) at RT 2.45 min *m*/*z* 566.3491, full scan mass spectrum (**I**), tandem mass spectrum (**J**). (**K**) The pathway analysis visualized by a bubble plot. (**L**) Heatmap and hierarchical cluster analysis of differential metabolites. (**M**) The mRNA levels of genes related to sphingolipid metabolism. TG: triglyceride, T-CHO: total cholesterol, PCA: principal component analysis, OPLS-DA: Orthogonal Partial Least Squares-Discriminant Analysis. * *p* < 0.05, ** *p* < 0.01, and *** *p* < 0.001.

**Figure 3 nutrients-16-02010-f003:**
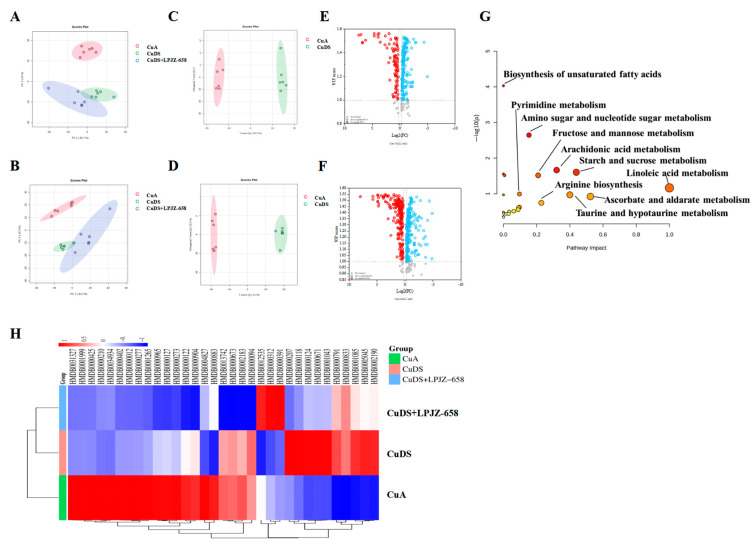
Effect of LPJZ-658 on serum metabolites in MASLD mice. PCA score plots of the annotated metabolites ((**A**) ESI+ and (**B**) ESI−). OPLS-DA score plots of identified metabolites ((**C**) ESI+ and (**D**) ESI−). Volcano plots of identified metabolites ((**E**) ESI+ and (**F**) ESI−). (**G**) The pathway analysis visualized by a bubble plot. (**H**) Heatmap and hierarchical cluster analysis of differential metabolites between. PCA: principal component analysis, OPLS-DA: Orthogonal Partial Least Squares-Discriminant Analysis.

**Figure 4 nutrients-16-02010-f004:**
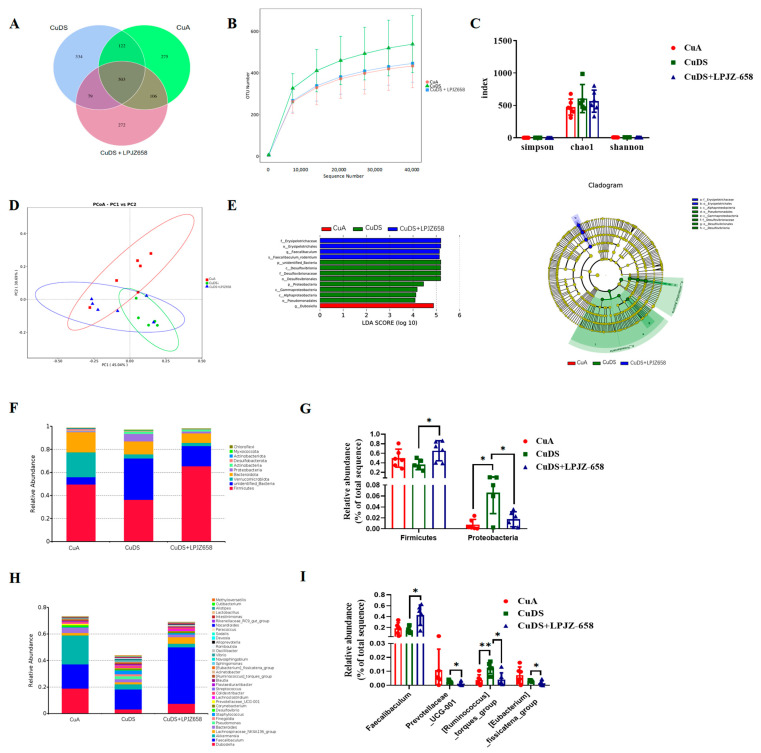
Effect of LPJZ658 on the composition of gut microbiota: (**A**) Venn diagram. (**B**) Rarefaction curves of cecal contents samples. (**C**) Alpha diversity indices. (**D**) PCoA plots (Bray-Curtis distance). (**E**) The LEfSe analysis of differential microbial taxa: distribution histogram of LDA values (left); evolutionary branching diagram (right). The relative abundance of cecal contents bacterial phyla (**F**) and general. (**H**) The differential microbiota at the phylum (**G**) and genus (**I**) level. PCoA: principal coordinate analysis, LEfSe: linear discriminant analysis effect size, LDA: linear discriminant analysis. * *p* < 0.05, ** *p* < 0.01.

**Figure 5 nutrients-16-02010-f005:**
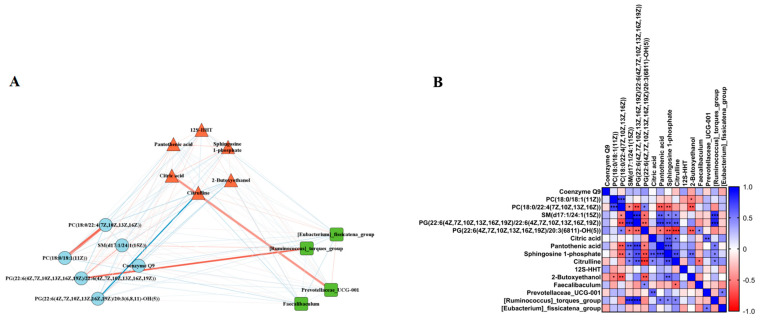
Correlation analysis. (**A**) Correlation network diagram between differential gut microbiota, serum metabolites, and liver lipidomic. Thicker lines indicate a higher correlation. The red line indicates a positive correlation, while the blue line indicates a negative correlation. (**B**) Correlation between differential gut microbiota, serum metabolites, and liver lipidomic. * *p* < 0.05, ** *p* < 0.01, and *** *p* < 0.001.

## Data Availability

All data generated or analyzed during this study are included in this published article.

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
