# Peer review of "Exploring the Regulatory Effect of LPJZ-658 on Copper Deficiency Combined with Sugar-Induced MASLD in Middle-Aged Mice Based on Multi-Omics Analysis"

_nutrients, 2024, doi:10.3390/nu16132010_

Round 1

Reviewer 1 Report

Comments and Suggestions for Authors

Dear Editor, dear authors,

The manuscript entitled "Exploring the Regulatory Effect of LPJZ-658 on Copper Deficiency Combined with Sugar-Induced NAFLD in Meddle-Aged Mice Based on Multi-Omics Analysis" is an original manuscript reporting the role of Lactiplantibacillus planta-20 rum LPJZ-658 on NAFLD. The authors use a model of sugar intake with copper deficiency to evaluate the role of the probiotic strain. Also, they use several state-of-the-art methods to evaluate metabolic changes in the liver and serum, as well as the microbiota. However, I have significant concerns regarding the model. Although this model was used in other publications, it does not render NAFLD. Liver weight was not changed, liver lipid content was only modestly increased and the histological analysis shows some localised lipid droplets, but far from being NAFLD. At most, this is a model of hyperlipidemia with modest fat deposition in the liver.

Another major concern about this study is the fact that the authors do not describe the data obtained. They only describe the experiments performed, but not the results obtained. Most of the results section describes alterations in the tissue and serum metabolome but the authors don’t actually describe the results obtained in detail. For instance, the section:

To determine whether LPJZ-658-induced changes in liver lipid metabolome and serum metabolome were associated with the effects of gut microbiota, we selected differential metabolites and genus-level colony abundance among the three groups for Spearman correlation analysis. As shown in Figure 5A-B, We found significant correlations between LPJZ-658-induced lipid metabolomics and serum metabolomics metabolites and genus-level gut microbiota abundance.

Here, the authors mention that they obtained correlations but don’t even describe which correlations were obtained.

Overall, the manuscript is somehow confusing because the authors use innovative methods, an animal model which has not NAFLD, and describe their data in a very poor manner.

Other major concerns:

-          Bar plots are not the most suitable way to show data. Authors should change to dots with mean and SEM, instead of bars. Or at least show both. Also, the number of replicates should be mentioned in the figure legends

-          English needs revision, some parts are difficult to read.

Comments on the Quality of English Language

English should be improved, in some parts of the text is difficult to understand

Author Response

We would like to express our gratitude for the opportunity to revise our manuscript entitled “Exploring the Regulatory Effect of LPJZ-658 on Copper Deficiency Combined with Sugar-Induced NAFLD in Meddle-Aged Mice Based on Multi-Omics Analysis” (Manuscript ID: nutrients-3020293). We sincerely thank all the reviewers for their valuable feedback, which we have carefully considered and used to improve the quality of our work.

We have addressed the reviewers’ comments in a point-by-point manner below, and we have made the necessary revisions to the manuscript. Specific concerns raised by the reviewers have been numbered for clarity. Our responses are given in normal font, and changes/additions to the manuscript are highlighted in red.

Thank you again for your time and consideration.

Response to the comments of Reviewer 1

Point 1: Although this model was used in other publications, it does not render NAFLD. Liver weight was not changed, liver lipid content was only modestly increased and the histological analysis shows some localised lipid droplets, but far from being NAFLD. At most, this is a model of hyperlipidemia with modest fat deposition in the liver.

Response 1: We feel great thanks for your professional review work on our article. It is true as the Reviewer considers, that Non-alcoholic fatty liver disease (NAFLD) ranges from simple steatosis and nonalcoholic steatohepatitis (NASH) with variable degrees of fibrosis to cirrhosis. In this study, we used a copper deficiency diet and combined it with a sugar solution-induced mouse model. Although there were no significant differences in the liver/body weight ratios between any of the groups. But compared with the CuA group, the serum levels of T-CHO, LDL, and NEFA were significantly elevated and the HDL levels were significantly lower in the CuDS group. In addition, the serum AST levels were significantly increased in the CuDS group. Furthermore, the histopathological examination of the liver demonstrated that the CuDS group enhanced hepatic lipid droplets and TG content compared with the CuA group. These results have identified that copper deficiency combined with a high-sugar diet-induced hyperlipidemia and modest fat deposition in middle-aged male mice. Of note, copper status is affected by age, gender, and hormone. Plasma copper concentrations and ceruloplasmin levels were higher in women than in men [1,2]. Moreover, previous studies have shown sex differences in the alterations of hepatic steatosis in response to dietary copper-fructose interactions in rats [3]. This suggests to us that the degree of NAFLD induced in the male middle-aged mice used in this study may be influenced by age and sex, and thus further research is still needed to clarify identified.

Point 2: Another major concern about this study is the fact that the authors do not describe the data obtained. They only describe the experiments performed, but not the results obtained. Most of the results section describes alterations in the tissue and serum metabolome but the authors don’t actually describe the results obtained in detail. For instance, the section:

“To determine whether LPJZ-658-induced changes in liver lipid metabolome and serum metabolome were associated with the effects of gut microbiota, we selected differential metabolites and genus-level colony abundance among the three groups for Spearman correlation analysis. As shown in Figure 5A-B, We found significant correlations between LPJZ-658-induced lipid metabolomics and serum metabolomics metabolites and genus-level gut microbiota abundance.”

Here, the authors mention that they obtained correlations but don’t even describe which correlations were obtained.

Response 2: We sincerely appreciate the valuable comments. We have added a description of the correlations obtained from the data.

Point 3: Bar plots are not the most suitable way to show data. Authors should change to dots with mean and SEM, instead of bars. Or at least show both. Also, the number of replicates should be mentioned in the figure legends.

Response 3: Considering the Reviewer’s suggestion, we changed the representation of the data to a dot plot. In addition, we added a description of the number of experimental repetitions in “2.2 Animal experimental design and sample collection”.

Point 4: English needs revision, some parts are difficult to read.

Response 4: We apologize for the poor language of our manuscript. We worked on the manuscript for a long time and the repeated addition and removal of sentences and sections led to poor readability. We tried our best to improve the manuscript and made some changes to the manuscript and have also involved native English speakers for language corrections. We hope that the flow and language level have been substantially improved. And we hope the revised manuscript will be acceptable to you.

According to the reviewer’s comments, we have revised the manuscript. If there are any other modifications we could make, we would like very much to modify them and we really appreciate your help. We appreciate for your warm work earnestly, and hope the correction will meet with approval. Once again, thank you very much for your comments and suggestions.

Best Wishes to you!

Yours sincerely,

Cuiqing Zhao

References

  1. Milne, D.B.; Johnson, P.E. Assessment of copper status: effect of age and gender on reference ranges in healthy adults. Clin Chem 1993, 39, 883-887.
  2. Millo, H.; Werman, M.J. Hepatic fructose-metabolizing enzymes and related metabolites: role of dietary copper and gender. J Nutr Biochem 2000, 11, 374-381, doi:10.1016/s0955-2863(00)00093-0.
  3. Song, M.; Yuan, F.; Li, X.; Ma, X.; Yin, X.; Rouchka, E.C.; Zhang, X.; Deng, Z.; Prough, R.A.; McClain, C.J. Analysis of sex differences in dietary copper-fructose interaction-induced alterations of gut microbial activity in relation to hepatic steatosis. Biol Sex Differ 2021, 12, 3, doi:10.1186/s13293-020-00346-z.

Reviewer 2 Report

Comments and Suggestions for Authors

In the reviewed article, the authors included research results on the influence of copper ions and the LPJZ-658 microbiome on liver disease. When reading it, it is hard not to get the impression that it was not fully thought out and prepared with due care.

The title itself raises doubts. Using the word sugar since it only refers to fructose does not seem appropriate. Rather, the term "fructose-induced NAFLD" should be used, as it appears later in the text - line 331.

It seems that the text should be checked for English and typing errors should be removed.

Line 20 - The sentence seems unfinished.

Line 128 - The word "The" should probably be lowercase.

Line 184 - What did the authors mean by writing [M-H]-?

There is no summary of the text in the form of a Conclusions.

The Supplementary Materials are written very poorly, contain many errors and absolutely need to be corrected.

In chemical formulas, numbers should be written in subscript - H2O, HCOONH4.

When writing Kv, did the authors have kV in mind?

The authors are inconsistent in writing numbers and units. Sometimes they write it together, and then separately.

It seems that there is no point in writing that the samples were freeze-dried with a freeze dryer.

Does such an entry make sense - Mobile phase A was 0.1% formic acid in water (Phase A)...?

Comments on the Quality of English Language

The text is difficult to read and contains illogical sentences. In my opinion, the authors should ask someone more fluent in English to help improve it.

Author Response

We would like to express our gratitude for the opportunity to revise our manuscript entitled “Exploring the Regulatory Effect of LPJZ-658 on Copper Deficiency Combined with Sugar-Induced NAFLD in Meddle-Aged Mice Based on Multi-Omics Analysis” (Manuscript ID: nutrients-3020293). We sincerely thank all the reviewers for their valuable feedback, which we have carefully considered and used to improve the quality of our work.

We have addressed the reviewers’ comments in a point-by-point manner below, and we have made the necessary revisions to the manuscript. Specific concerns raised by the reviewers have been numbered for clarity. Our responses are given in normal font, and changes/additions to the manuscript are highlighted in red.

Thank you again for your time and consideration.

Response to the comments of Reviewer 2

Point 1: Using the word sugar since it only refers to fructose does not seem appropriate. Rather, the term "fructose-induced NAFLD" should be used, as it appears later in the text - line 331.

Response 1: We apologize for our inaccurate representation. We have made correction according to the Reviewer’s comments.

Point 2: It seems that the text should be checked for English and typing errors should be removed.

Response 2: We apologize for the poor language of our manuscript. We worked on the manuscript for a long time and the repeated addition and removal of sentences and sections led to poor readability. We tried our best to improve the manuscript and made some changes to the manuscript and have also involved native English speakers for language corrections. We hope that the flow and language level have been substantially improved. And we hope the revised manuscript will be acceptable to you.

Point 3: Line 20 - The sentence seems unfinished.

Response 3: We are sorry for our limited description. We have made modifications to this sentence.

Point 4: Line 128 - The word "The" should probably be lowercase.

Response 4: We are very sorry for our carelessness. We have corrected the word “The” in the revised manuscript.

Point 5: Line 184 - What did the authors mean by writing [M-H]-?

Response 5: [M-H]- means the Adduct ion of serum or liver samples in negative ionization mode [1,2].

Point 6: There is no summary of the text in the form of a Conclusions.

Response 6: We sincerely appreciate the valuable comments. We have summary the manuscript in the form of “Conclusions” according to the Reviewer’s suggestion.

Point 7: The Supplementary Materials are written very poorly, contain many errors and absolutely need to be corrected.

Response 7: We sincerely appreciate the valuable comments. We have corrected “Supplementary Materials” and have also involved native English speakers for language corrections according to the Reviewer’s suggestion.

Point 8: In chemical formulas, numbers should be written in subscript - H2O, HCOONH4.

Response 8: We sincerely thank you for careful reading. We have corrected the chemical formulas.

Point 9: When writing Kv, did the authors have kV in mind?

Response 9: We sincerely thank you for careful reading. According to your suggestion, we have corrected the rule of voltage units representation.

Point 10: The authors are inconsistent in writing numbers and units. Sometimes they write it together, and then separately.

Response 10: We feel sorry for our carelessness. In our resubmitted manuscript, we have standardized the writing of numbers and units. Thanks for your correction.

Point 11: It seems that there is no point in writing that the samples were freeze-dried with a freeze dryer.

Response 11: We feel great thanks for your professional review work on our article. In fact, in this study, we freeze-dried the liver and serum samples to increase the concentration of the samples.

Point 12: Does such an entry make sense - Mobile phase A was 0.1% formic acid in water (Phase A)...?

Response 12: We feel sorry for our inaccurate description. We have corrected this sentence. Thanks for your correction.

According to the reviewer’s comments, we have revised the manuscript. If there are any other modifications we could make, we would like very much to modify them and we really appreciate your help. We appreciate for your warm work earnestly, and hope the correction will meet with approval. Once again, thank you very much for your comments and suggestions.

Best Wishes to you!

Yours sincerely,

Cuiqing Zhao

References

  1. Yin, X.L.; Peng, Z.X.; Pan, Y.; Lv, Y.; Long, W.; Gu, H.W.; Fu, H.; She, Y. UHPLC-QTOF-MS-based untargeted metabolomic authentication of Chinese red wines according to their grape varieties. Food Res Int 2024, 178, 113923, doi:10.1016/j.foodres.2023.113923.
  2. Wang, Y.; Guo, W.; Xie, S.; Liu, Y.; Xu, D.; Chen, G.; Xu, Y. Multi-omics analysis of brain tissue metabolome and proteome reveals the protective effect of gross saponins of Tribulus terrestris L. fruit against ischemic stroke in rat. J Ethnopharmacol 2021, 278, 114280, doi:10.1016/j.jep.2021.114280.

Round 2

Reviewer 2 Report

Comments and Suggestions for Authors

The revised version of the manuscript sent by the authors contains the most important corrections I suggested. The authors also included short conclusions in the form of a separate chapter. The additional materials have also been written more carefully. The text contains fewer typing errors and its form is closer to the high standards of the journal.

Author Response

Thank you for your comment, if there are any other modifications we could make, we would like very much to modify them and we appreciate your help. We appreciate your warm work earnestly. Once again, thank you very much for your comments and suggestions.

Best Wishes to you!

Yours sincerely,

Cuiqing Zhao